# The Effect of Different Electric Toothbrush Technologies on Interdental Plaque Removal: A Systematic Review with a Meta-Analysis

**DOI:** 10.3390/healthcare12101035

**Published:** 2024-05-16

**Authors:** Robert David Lewis, Shalini Kanagasingam, Neil Cook, Marta Krysmann, Kathryn Taylor, Flavio Pisani

**Affiliations:** School of Medicine and Dentistry, University of Central Lancashire, Preston PR1 2HE, UK; rdlewis@uclan.ac.uk (R.D.L.); skanagasingam1@uclan.ac.uk (S.K.); ncook2@uclan.ac.uk (N.C.); mkrysmann@uclan.ac.uk (M.K.); ktaylor40@uclan.ac.uk (K.T.)

**Keywords:** interproximal plaque, oscillating–rotating, sonic, electric toothbrush, Rustogi modified naval plaque index, Turesky modification of the Quigley–Hein index

## Abstract

The removal of dental plaque from the gingival margins of the teeth is essential to maintaining periodontal health. Whilst it has been established that electric toothbrushes demonstrate a greater ability to remove plaque, no systematic review has specifically investigated which technology is better for removing plaque from the interdental tooth surfaces, where plaque control may be more difficult. Three databases were searched until October 2023: MEDLINE and DOSSS via EBSCOhost and Embase. Data extraction was carried out on studies which met the inclusion criteria, and a risk of bias assessment was completed. The study findings were combined via a narrative synthesis and a meta-analysis where appropriate. A total of 77 studies were found, out of which 14 were selected and included in the analysis. The mean difference in interproximal plaque reduction, measured using the Rustogi Modified Naval Plaque Index (RMNPI) at 8 weeks, was 0.09 (*p* < 0.00001) in favor of the oscillating–rotating toothbrush. At 6 and 12 weeks, the mean difference in plaque reduction (RMNPI) was 0.05 (*p* = 0.0008) and 0.04 (*p* = 0.0001) in favor of the oscillating–rotating toothbrush, respectively. The studies show a tendency for oscillating–rotating toothbrushes to remove more interproximal plaque than oscillating toothbrushes, especially in a short time (8 weeks).

## 1. Introduction

The association between dental plaque, gingivitis and periodontitis has long been established [1,2]. Dental plaque contains bacteria which irritate the gingival tissues, causing them to become inflamed, leading to gingivitis. If left to progress, gingivitis can lead to the breakdown of the periodontium [3,4]. Furthermore, periodontitis has been linked with other systemic chronic inflammatory conditions, meaning its prevention can have a positive impact on overall health as well as oral health [4]. Lang et al. [5] demonstrated that plaque formation begins at the interdental sites. Despite this, Marchesan et al. [6] found that nearly one-third of respondents based in the United States reported that they do not use any form of interdental cleaning aid. This study also found that those who did report interdental cleaning as part of their oral hygiene routine at home were more likely to exhibit periodontal health. It is clear, therefore, that whilst the removal of interdental plaque is important in the prevention of diseases of the periodontium, interdental cleaning is not routinely carried out by a sizeable proportion of patients [7,8]. If interdental cleaning is not ubiquitous within the population, advocating for the type of toothbrush which is most effective in removing interdental plaque may go some way towards addressing this issue, ensuring that patients who are perhaps reluctant or have low motivation to carry out designated interdental plaque removal still reap the maximum benefit from the toothbrush they use.

A consensus report by Chapple et al. [4] concluded that mechanical plaque removal is recommended to maintain gingival health and that the evidence base suggests that electric toothbrushes demonstrate a greater reduction in plaque than manual brushes both in the short (up to 3 months) and long term (over 3 months), which is also reflected by a reduction in gingival inflammation. This consensus report is supported by the findings of a Cochrane review [9], which again supports the use of electric toothbrushes over manual toothbrushes for plaque reduction. However, the analysis did not directly compare different electric toothbrush types with one another, despite their different modes of action and required brushing techniques. Patients often look to their dental clinician for guidance when it comes to oral hygiene, and such a comparison may help to guide clinical recommendations as to which type of tool should be used by patients in a more performant way. Whilst it can be concluded from this review that electric toothbrushes are more effective than manual toothbrushes in removing plaque, directly comparing different electric toothbrush modalities to one another may provide clinicians with more certainty when providing toothbrush recommendations to their patients, should one modality demonstrate a greater cleaning ability. A recent systematic review [10] directly compared different electric toothbrush modalities. The authors concluded that oscillating–rotating electric toothbrushes were the most efficacious at removing plaque when compared to manual and sonic toothbrushes, but the review did not investigate their effects on interproximal plaque reduction, despite the interproximal sites being highly at risk of plaque formation [5]. Thomassen et al. [11] also compared manual, oscillating–rotating and sonic electric toothbrushes and found a marginal benefit in plaque removal with the oscillating–rotating brush compared to the other two modalities. Again, however, this study makes no specific reference to interproximal plaque reduction.

Zou et al. [12] compared different electric toothbrush modalities in their systematic review, which found that oscillating–rotating electric toothbrushes removed statistically significantly more interproximal plaque than both manual and sonic toothbrushes. However, this study was not independently carried out, as the authors are affiliated with the manufacturer of an oscillating–rotating toothbrush.

To date, there has been no independent systematic review published which focuses specifically on the efficacy of interproximal plaque reduction using different electric toothbrush modalities. The present systematic review aims to address this.

The research question of the present review was “Does the use of an oscillating–rotating electric toothbrushing remove more interdental plaque compared to an oscillating (sonic) electric toothbrushing in adults?”.

## 2. Materials and Methods

### 2.1. Study Registration

The current systematic review was registered on PROSPERO on 5 April 2022, with the number CRD42022319371.

### 2.2. Literature Search

Following the recommendations of the Preferred Reporting Items for Systematic Reviews and Meta-Analyses (PRISMA) statement [13], a PICOS framework was used to develop and define the research question:

P (Population): Adults performing oral hygiene.

I (Intervention): Any electric toothbrush in which the bristles move according to oscillating–rotating action.

C (Comparator): Any electric toothbrush which employs an oscillating “side-to-side” movement, with no rotational component, often termed “sonic” toothbrushes.

O (Outcomes): Interproximal plaque index variation from baseline examination.

S (Study design): Randomised controlled trials only.

Three databases were searched for relevant articles in January 2022: MEDLINE and Dentistry and Oral Sciences Source (via EBSCOhost) and Embase (via Ovid). 

The searches were carried out and adapted as appropriate for each database (Table 1). Boolean operators were used to combine the searches. Each individual PICO component was combined with OR. The final search combined the PICO terms with AND. A manual search was carried out on field-related journals such as *Journal of Periodontology*, *Journal of Clinical Periodontology*, *International Journal of Dental Hygiene* and *the American Journal of Dentistry*. There are two commonly used brands of electric toothbrush which utilise either an oscillating–rotating or sonic mode of action, as can be seen from Amazon’s “Best Sellers in Electric Toothbrushes” webpage [14]. Representatives from these two brands, namely Philips and Oral-B, were contacted by email to see whether they could provide any gray literature which may address the research question.

### 2.3. Literature Selection

The eligibility criteria were set out as follows:

#### 2.3.1. Inclusion Criteria

The following table (Table 2) shows the inclusion criteria of the studies.

The included studies had to have been published from the year 2000 onwards, as it is reasonable to assume that electric toothbrush technology has changed considerably in that time [15], and any conclusions drawn from this review should be based on the electric toothbrush technologies available to the public today.

#### 2.3.2. Exclusion Criteria

Table 3 shows the exclusion criteria for the present review.

### 2.4. Literature Screening

The studies generated by the search were first screened by title by two of the authors (R.L. and F.P.). Relevant studies from the database searches were imported into RefWorks [16]. Duplicates were excluded, and the studies were then checked by the same two authors through two levels of screening (titles and abstracts) and the inter-agreement score was recorded according to Cohen’s Kappa score to check the consistency [17] (Appendix A). “Maybes” were not counted as disagreements. Any disagreement was discussed and, where appropriate, moderated by a third reviewer (N.C.). The final searches were all completed in October 2023. The results were summarised into a PRISMA flowchart (Figure 1) [13].

### 2.5. Outcome Measures

The outcome measure of interest was the reduction in the reported interproximal plaque level using the Rustogi Modified Naval Plaque Index (RMNPI). This index extends the scoring of plaque to the interproximal (mesial and distal) tooth areas and to the gumline (marginal gingival) region, as well as the total tooth. It divides the buccal and lingual surfaces into nine areas, which are scored on the presence or absence of dental plaque [19]. Alternative plaque indices were also considered. Turesky modification of the Quigley–Hein index (TMQH) considers the amount of the tooth surface covered by plaque as a fraction of the total tooth surface and splits the buccal and lingual tooth surfaces into mesial, distal and mid [19]. The wet weight of interproximal plaque was also used as an outcome measure.

### 2.6. Data Extraction

A data extraction table was used for each study so that relevant information could be laid out in an accessible format for later comparison (Table 4). Development of the data extraction table was an iterative process based on the findings from the first few studies, which were screened to ensure that all relevant data were extracted. Of particular interest were the study design, outcome measures and the results on the interproximal plaque reduction. The data extraction table was based on the Cochrane guidelines [20].

### 2.7. Assessment of Heterogeneity

A thorough assessment of the heterogeneity across the included studies was detailed according to the following factors:
(a)Study design,(b)Trial duration and evaluation timeline,(c)Population sample and demographics,(d)Baseline oral health features across the test and control groups,(e)Smoking habits,(f)Previous use of manual or electric toothbrushes.

### 2.8. Quality Assessment

Once the data extraction was completed, the papers were critically appraised using the Cochrane Risk of Bias 2 (RoB 2) tool, which was developed to allow for a more standardised way of assessing bias compared to its predecessor [35] and the Jadad tool [36].

RoB 2’s methodological quality was evaluated according to the five risk of bias domains of randomisation, deviation from the intended intervention, missing outcome data, measurement of the outcome and the selection of the reported result. The overall and domain-related judgement for each included paper was reported in tabular format, displaying the risk of bias in different colors (red for high-risk, amber for some concerns and green for low-risk) according to the Cochrane RoB 2 resources tool [37]. The Jadad tool is a numeric scale based on three items: randomisation, blinding and drop-out reporting. The appraisal was carried out independently by two reviewers (R.D.L. and F.P.) and measured according to Cohen’s Kappa score to check the consistency. It was agreed to include the studies at moderate risk of bias in the qualitative and quantitative syntheses due to the expected small number of retrievable studies.

### 2.9. Data Synthesis and Publication Bias Check

Following data extraction and critical appraisal, the included papers were analysed by way of narrative synthesis. To guide the narrative synthesis process, the flowchart by Rodgers et al. [38] was consulted, which provides guidance on how to synthesise the data, look for patterns within them, establish the strength of these patterns and then make conclusions based on these patterns.

To identify any potential publication bias, a funnel plot analysis was performed [39].

Where appropriate, dependent on the studies having similar characteristics and methodologies, a meta-analysis was carried out on the outcome data, displayed as the Weighted Mean Difference (WMD). For continuous variables, mean differences and 95% confidence intervals were used to summarise the data for each study. Forest plots and funnel plots were created to illustrate the effects of different studies and the overall average results. Review Manager (RevMan) version 5.3. for macOs from Cochrane collaboration was used for all the analyses. Statistical significance was set as a *p* value < 0.05.

The statistical heterogeneity among the studies was assessed using the I^2^ test [40]. A random-effects model was adopted as per the hypothesis of a population of studies with possible variations.

## 3. Results

### 3.1. Study Selection

From an initial batch of 156 articles retrieved via databases and registers, 90 were removed as duplicates. Eleven studies were yielded from the manual search and from a recently published systematic review [10].

Figure 1 shows a PRISMA flowchart [18] which documents the transition from the initial results to the final ones for data extraction.

Seventeen studies were eligible for full-text review; however, four were not retrievable. The remaining 13 studies were retrieved, fully assessed and included. From the manual search, 6 out of 10 were available for the full-text assessment, of which all but 1 [29] were excluded, due to a lack of reporting on the interproximal plaque level (3/6), to the sonic toothbrush being multi-directional (1/6) or to the lack of a sonic toothbrush control group in the study (1/6). Fourteen studies were included at the end of the selection process.

To test the agreement in the process performance between the two reviewers (R.L. and F.P.), Cohen’s Kappa statistics were used at different stages of the screening [17]. The mean overall calculated score was 0.92, which indicated the substantial reliability of the overall procedure [41]. Disagreements arose when there was some ambiguity within the abstracts of the articles as to whether interproximal plaque was specifically reported on. Following discussion, where uncertainty remained, the papers were retrieved so that the full text could be analysed and either included or excluded as appropriate based on the aforementioned study selection criteria.

The extraction of data from each study was carried out by one supervised single author (R.L.) using a standardised tabular set. The format reflected the PICOS template, providing specific demographic data, the main features of the intervention and comparative groups and the main outcome characteristics in terms of the interproximal plaque variation from the baseline to the staged timelines as means (M) and standard deviation (SD)/standard error (SE). Once complete, the data ultimately suitable for the quantitative analysis were checked, grouped by time and processed with the use of statistical software (RevMan 5.3). Table 4 shows the study characteristics of the included studies.

### 3.2. Study Methodologies

Almost all the studies (10) described a stratification process for randomisation. One study [22] used a set of two Latin squares for randomisation, while four [27,29,30,31] did not clearly describe their process beyond stating that participants were randomised. 

The participant numbers within the studies ranged from 148 [31] to 45 [25]. Twelve of the studies assessed plaque reduction using the Rustogi Modified Naval Plaque Index (RMNPI). One study [26] used the Turesky modified Quigley–Hein plaque index (TMQH). One [21] used the wet weight of plaque to measure the reduction.

### 3.3. Risk of Bias

The Cochrane Risk of Bias 2 (RoB 2) tool [37] was used to assess each study and to assign the overall risk of bias, while the Jadad tool [36] was used as a reference. The results of this were depicted on a colour-coded chart for easy consultation (Table 5).

Overall, three of the studies [33,35,36] were deemed to be at moderate risk of bias due to their adherence to the intervention category, as the test and control groups differed in their use of apps or brushing guides to help ensure thorough toothbrushing. It is not clear to what extent these brushing aides affected the results between the test and control groups, but as the results fall in line with the results from most of the papers, a decision was made to score them as moderate and still include them in the analysis. 

All of the remaining studies were deemed to have a low risk of bias. Comparative analysis of the RoB 2 and Jadad tools helped to clarify whether the existence of reported dropouts would affect the quality of the study, as per the possible underpowered sample used by Cchauana Vasquez et al. [31].

To test the agreement in the process performance between the two reviewers (R.L. and F.P.), a Cohen’s Kappa statistical test was used to calculate the different quality assessments performed independently.

The mean calculated Kappa score was 0.57, which indicated the substantial discrete reliability of the overall procedure [41]. Disagreements in the risk of bias score were due to the subjective nature of the RoB 2 tool. They were resolved following discussion between the two reviewers until an agreement was made.

A decision was made to still include the studies with a moderate risk of bias and test them for heterogeneity where appropriate in the meta-analysis, thereby generating as large a dataset as possible from which to draw conclusions.

### 3.4. Categorisation of Studies

The included studies were categorised into different groups according to the follow-up interval with the aim of making them comparable for the quantitative meta-analysis. Four groups were available for the analysis at 6, 8, 12 and 24 weeks.

### 3.5. Qualitative Synthesis

The included trials were all randomised controlled trials, 6 with a crossover and 8 with a parallel design.

The earliest study was published in 2004 [21], and the most recent one was published in 2021 [34]. Most of the studies allowed the use of the test and control toothbrushes at home, although one did not [22]. The studies had a range of follow-up times, from four days to 24 weeks. Four of the studies provided results for 12 weeks from the baseline [33,34,35,38], which was the most common follow-up period. Two studies reported findings at 24 weeks [38,40], two at 8 weeks [37,39] and three at 6 weeks [33,35,36] from the baseline. All the studies used a single, blinded examiner, reducing the risk of examiner bias in the pre- and post-brushing plaque assessments. All of the papers used an Oral-B oscillating–rotating toothbrush as the oscillating–rotating brush, and the majority (12) of them used a Philips Sonic toothbrush (Sonicare). Only one study [28] did not include the Philips sonic electric toothbrush, instead using one made by Colgate with the same technology. One study [32] investigated the GEVILAN GET011 sonic toothbrush as well as the Philips Sonicare brush.

### 3.6. Interproximal Plaque Reduction

#### 3.6.1. Overall Comparison of OR TBs and Sonic TBs

Overall, most of the studies found that the oscillating–rotating electric toothbrush removed more interproximal dental plaque than the sonic toothbrush. Only two studies [27,38] found more beneficial effects from the sonic toothbrush. Based on the RMNPI as the interproximal plaque outcome measure, the largest reduction from the baseline for an oscillating–rotating toothbrush was found by Strate et al. [23] (mean RMNPI reduction 0.966 (SD = 0.06)). However, this study compared the test and control toothbrushes only once. The largest reduction over a more prolonged period was found by Biesbrock et al. [25], 12 days after the baseline assessment (mean RMNPI reduction 0.884 ± 0.013). For a sonic toothbrush, the largest reduction was found by Goyal et al. [34] (mean RMNPI reduction = 0.794 ± 0.0101) at just day 1 following the baseline assessment, although it was less than that of the oscillating–rotating brush, while the next highest reduction was found by Biesbrock et al. [25] at 12 days (mean RMNPI reduction = 0.724 ± 0.013).

#### 3.6.2. Time-Wise Comparison of OR TBs and Sonic TBs

In order to visualise the comparison, a time-wise analysis according to the different milestones has been reported (Table 6).

### 3.7. Quantitative Analysis

#### 3.7.1. Meta-Analysis

Where appropriate, the data were synthesised according to a meta-analysis (Table 7). A concomitant funnel plot (Figure 2) was created to check the eligibility of the comparative analysis. This demonstrated an overall low risk of publication bias.

The results of the meta-analyses (Table 7) show that at 6, 8 and 12 weeks, the oscillating–rotating electric toothbrushes removed more interproximal plaque (RMNPI) than the sonic toothbrushes. The difference was greatest at eight weeks. No statistically significant difference was established at six months.

#### 3.7.2. Heterogeneity Assessment

The I^2^ statistics showed substantial heterogeneity at the majority of the follow-up time periods.

Appropriate studies were grouped by follow-up time. As can be seen from the forest plots (Figure 3), the results were found to be in favour of the oscillating–rotating toothbrush in most of the studies.

## 4. Discussion

The results of this systematic review demonstrate that the majority of studies which address the research question have found an oscillating–rotating toothbrush to be more effective than a sonic toothbrush in removing interproximal plaque. This effect tends to be greatest in the shorter term, with the highest plaque reduction for the oscillating–rotating brush at four days following the baseline [23] and the highest for the sonic brush at one day following the baseline [34]. However, the findings were generally consistent across all time periods. At all time periods, the difference in the interproximal RMNPI reduction between the two toothbrushes was low.

It may be that the bristles of the oscillating–rotating toothbrush work best in the short term when the patients’ technique is better and they become less effective in these areas if the patients show a tendency to “scrub” their teeth with the brush over the longer term, preventing the bristles from adequately accessing the interproximal areas.

The RMNPI and TMQH have previously been compared and were both demonstrated to be adequate for assessing plaque reduction differences between different toothbrushes [19]. Weighing plaque to measure this reduction has been shown to be less performant in demonstrating the differences between pre- and post-brushing plaque levels when compared to a plaque index [42].

The meta-analyses were generally found to be in favour of an oscillating–rotating brush. However, there were a limited number of studies available for inclusion in each analysis, and only the six-week meta-analysis demonstrated an I^2^ value below 50%, suggesting some heterogeneity between the studies. This limits the reliability of the present meta-analysis for providing firm conclusions as to which electric toothbrush modality is best. Furthermore, the results of the meta-analyses, whilst statistically significant, may not represent clinically important reductions in interproximal plaque levels. Whilst the meta-analysis does show a tendency towards an increased interproximal plaque reduction with oscillating–rotating toothbrushes, firmer conclusions cannot be made without more homogeneous data and longer-term studies which may demonstrate more clinical significance.

This systematic review agrees with a Cochrane review which concluded that oscillating–rotating toothbrushes had the largest amount of research to support their plaque removal ability compared to other electric toothbrush designs [9]. It also supports the conclusion made by Grender, Adam and Zou [10] that oscillating–rotating toothbrushes are superior at removing plaque to sonic toothbrushes.

Three of the fourteen included studies were found to be of moderate risk of bias, while the remainder were all at low risk. There appears to be no available literature on this topic which is not supported in some way by a toothbrush manufacturer, and it must be recognised that any results published are likely to favour the supporting company’s toothbrush. The low number of published studies supported by sonic toothbrush manufacturers may suggest, however, that there have been few carried out which demonstrate that sonic brushes are better. This may also be the reason the studies which supported sonic toothbrushes used a different plaque index compared to all the others. Nevertheless, the funnel plots (Figure 2) which were generated to check for the existence of any publication bias suggested that there was not any in the present systematic review and analysis. Moreover, Heinemann et al. [43] argued that industry-sponsored studies must meet more rigorous standards than purely academic studies and that authors of both types of studies are still compelled to publish data which demonstrate statistically significant differences between the test and control groups. The quality check and the parallel use of Jadad with RoB 2 are effectively supportive of strict adherence to the study structure and results.

It was often not clear how similar the test and control groups were at baseline, and so there is the possibility that confounding factors may have accounted for the results if the two groups were not adequately similar. Participants may have even been more familiar with one of the toothbrushes being investigated, which could have led to an unfair advantage if they already felt comfortable using one toothbrush over another.

Another element to reflect on was the use of dedicated apps supporting patients in efficient toothbrushing. It is unclear how much the toothbrush mechanism, and not the effectiveness of the apps, was responsible for the differences in interproximal plaque reduction.

The recruitment process of many of the studies was unclear. The majority were carried out at a research facility, but there is no mention as to who the participants were or whether they received any financial reward for their participation in the studies. Of particular concern were the two papers by Klukowska et al. [35,36], which both reported data at six weeks from the baseline. It is not clear whether the studies represent entirely different study populations or whether there is some overlap. This is a problem, as it could mean that one study finding is then represented twice in the narrative synthesis and meta-analysis.

There are a number of limitations to this systematic review, as follows:There were a limited number of studies suitable for meta-analysis.There was considerable heterogeneity between the studies.Some of the studies were found to be of some concern for risk of bias.The longest follow-up period was six months, meaning there is a lack of long-term data available on this subject. Those studies with shorter follow-up periods are of limited value to this review, as their validity over a longer period was not established. Despite this, their findings remain in agreement with most of the longer-term studies.

Due to the widespread use of electric toothbrushes and the improvement of their technology, there is potential for future research to be carried out by independent researchers to compare these two modalities, so that the data can be published no matter which type is shown to be better. Longer-term studies could also be carried out, perhaps assessing plaque reduction up to a year from the baseline and with only a small number of assessments in between, which may be a better reflection of how effective the “real-world” interproximal plaque reduction using either toothbrush type would be.

Despite the limitations discussed above, there is evidence to suggest that oscillating–rotating toothbrushes are the more effective toothbrush type for removing interproximal plaque, as demonstrated by 12 of the 14 studies. This means that clinical recommendations based on the findings of this review can be made with some confidence, which, in turn, will help patients to make an informed choice when purchasing a toothbrush and will hopefully lead to a greater reduction in interproximal plaque for them, facilitating the maintenance of gingival health.

## 5. Conclusions

Oscillating–rotating electric toothbrushes appear to be more effective at removing dental plaque from the interproximal areas of teeth than sonic toothbrushes when used by adults. Across the relevant literature available on this topic, the tested oscillating–rotating toothbrushes demonstrated a tendency towards an increased reduction in interproximal plaque from the baseline compared to the sonic toothbrushes. This may be due to the generally smaller toothbrush head size allowing for better bristle placement between the teeth or due to the technique for using an oscillating–rotating brush being more easily learned by the patients. However, this difference has only been demonstrated over a relatively short time period, and the increased interproximal plaque reductions are small. Further, independent studies would be useful, perhaps reporting on a longer-term follow-up of participants, to help to overcome this issue and allow firm recommendations to be made.

## Figures and Tables

**Figure 1 healthcare-12-01035-f001:**
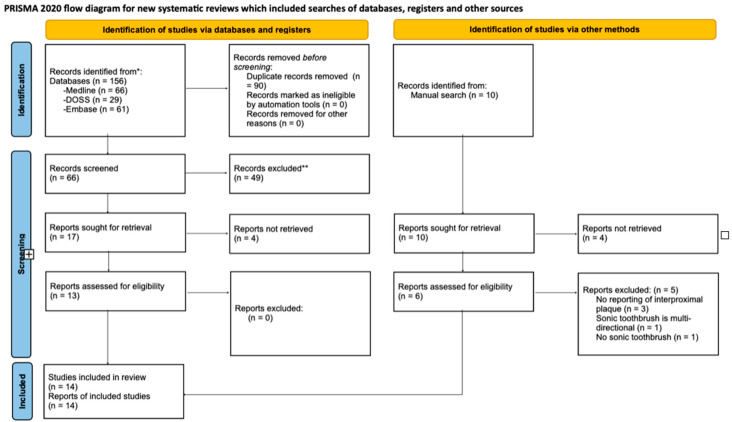
PRISMA flowchart with details of the screening stages [18]. * Consider, if feasible to do so, reporing the number of records identified from each database or register searched (rather than the total number across all databases/registers). ** If automation tools were used, indicate how many records were excluded by a human and how many were excluded by automation tools.

**Figure 2 healthcare-12-01035-f002:**
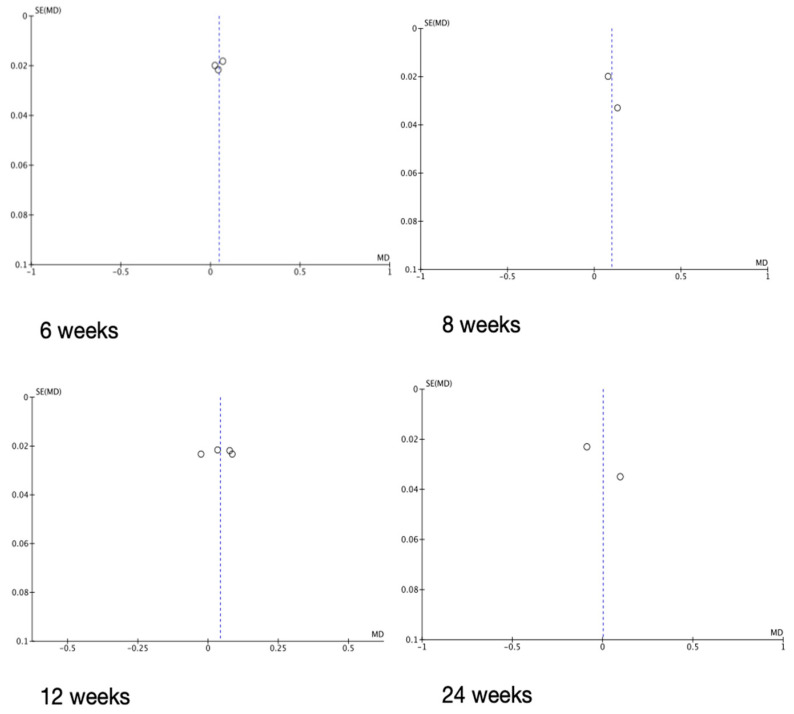
Funnel plots grouped by study follow-up intervals. The circles represent a single study. The *y*-axis shows standard error. The *x*-axis shows mean difference. The vertical dotted line represents the overall effect.

**Figure 3 healthcare-12-01035-f003:**
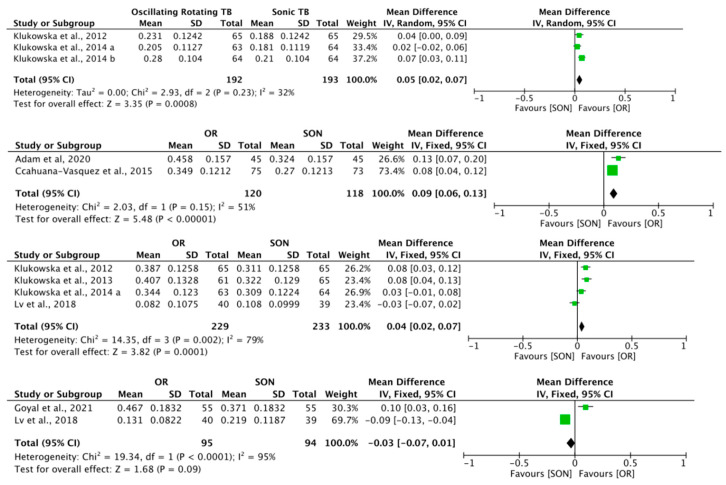
Forest plots of follow-up periods at 6, 8, 12 and 24 weeks. OR: oscillating–rotating TB; SON: sonic TB. Green boxes represent the study results. Horizontal line represents the 95% confidence interval. Diamonds represent the average of the studies within each subgroup [27,28,29,30,31,32,33,34].

**Table 1 healthcare-12-01035-t001:** Search strategy and terms. MEDLINE via EBSCOhost.

Main Concept	Search Number	Search Term	Boolean Operator
PROBLEM Interdental Plaque	S1	MH “Dental plaque”	
	S2	MH “Dental plaque index”	
	S3	“dental plaque”	
	S4	“dental plaque index”	
	S5	“plaque index”	
	S6	“dental biofilm”	
	S7	MH “Biofilms”	
	S8	biofilms	
	S9	MH “Dental Deposits”	
	S10	“Dental deposits”	
	S11	MH “toothbrushing”	
	S12	toothbrushing	
	S13	MH “oral hygiene”	
	S14	“oral hygiene”	
	S15		Combine S1–S14 with OR
	S16	interdental	
	S17	inter-dental	
	S18	interproximal	
	S19	inter-proximal	
	S20	proximal	
	S21	approximal	
	S22		Combine S16–S21 with OR
FINAL PROBLEM SEARCH	S23		Combine S15 and S22 with AND (search A)
INTERVENTION Toothbrushing with an oscillating toothbrush	S24	elect* N3 toothbrush*	
	S25	power* N3 toothbrush*	
	S26	battery N3 toothbrush*	
	S27		Combine S24–S26 with OR (search B)
	S28	oscillat*	
	S29	sonic	
	S30	MH “sonication”	
	S31	sonication	
	S32	“side-to-side”	
	S33	MH “Vibration”	
	S34	vibrat*	
	S35	Sonicare	
	S36		Combine S28–S35 with OR (search C)
FINAL INTERVENTION SEARCH	S37		Intervention final search = (search B) AND (search C)
COMPARISON Toothbrushing with an oscillating–rotating toothbrush	S38	oscillat* N3 rotat*	
	S39	Oral B	
	S40		Combine S39–S40 with OR (search D)
FINAL COMPARISON SEARCH	S41		Comparison final search = (search B) AND (search D)
FINAL SEARCH	S42		Final search = (Final problem search) AND (Final intervention search) AND (Final comparison search)

The search was adapted for the other databases. MEDLINE: Interdental plaque search terms: MH “Dental plaque”; MH “Dental plaque index”; “dental plaque”; “dental plaque index”; “plaque index”; “dental biofilm”; MH “Biofilms”; biofilms; MH “Dental Deposits”; “Dental deposits”; MH “toothbrushing”; toothbrushing; MH “oral hygiene”; “oral hygiene”; interdental; inter-dental; interproximal; inter-proximal; proximal; approximal. Electric toothbrush search terms: elect* N3 toothbrush*; power* N3 toothbrush*; battery N3 toothbrush*. oscillating toothbrush search terms: oscillat*; sonic; MH “sonication”; sonication; “side-to-side”; MH “Vibration”; vibrat*; Sonicare. Oscillating–rotating toothbrush search terms: oscillat* N3 rotat*; Oral-B. DOSS: Interdental plaque search terms: DE “Dental plaque”; “dental plaque”; “plaque index”; DE ”BIOFILMS”; “dental biofilm”; DE “DENTAL deposits”; “dental deposits”; toothbrushing; DE “ORAL hygiene”; “oral hygiene”; interdental; inter-dental; interproximal; inter-proximal; proximal; approximal. Electric toothbrush search terms: electr* N3 toothbrush*; power* N3 toothbrush*; battery N3 toothbrush*. Oscillating toothbrush search terms: DE “OSCILLATIONS”; oscillat*; sonic; DE “SONICATION”; sonication; DE “VIBRATION (Mechanics)”; vibrat*; Sonicare; “side-to-side”. Oscillating–rotating toothbrush search terms: oscillat* N3 rotat*; Oral-B. Embase: Interdental plaque search terms: tooth plaque/; “tooth plaque”; plaque index/; “plaque index”; “dental plaque”; biofilm/; biofilm; dental deposit/; “dental deposit”; tooth brushing/; toothbrushing; mouth hygiene/; “mouth hygiene”; “oral hygiene”; interdental; inter-dental; interproximal; inter-proximal; proximal; approximal. Electric toothbrush search terms: electr* adj3 toothbrush*; power* adj3 toothbrush*; battery adj3 toothbrush*. Oscillating toothbrush search terms: oscillation/; oscillat*; sonic; Sonication; vibration/; vibrat*; Sonicare; “side-to-side”. Oscillating–rotating toothbrush search terms: oscillat* adj3 rotat*; Oral-B.

**Table 2 healthcare-12-01035-t002:** Inclusion criteria of the studies.

Inclusion Criteria
Adult participants (18 years of age or older)
Studies measuring a change in interproximal plaque, measured using the Rustogi Modified Naval Plaque Index (RMNPI), Turesky modification of the Quigley–Hein index (TMQH) or wet weight of plaque.
Studies investigating self-performed oral hygiene
Studies with a randomised controlled trial design
Studies published since the year 2000

**Table 3 healthcare-12-01035-t003:** Exclusion criteria for the present review.

Exclusion Criteria
Children (under 18 years of age)
Participants with fixed or removable orthodontic appliances
Participants with removable oral prostheses, such as a denture
Participants unable to carry out self-performed oral hygiene
Studies not written or translated into English
Studies which are not randomised controlled trials

**Table 4 healthcare-12-01035-t004:** Study characteristics and results. OR, oscillating–rotating; RMNPI, Rustogi Modified Naval Plaque Index; MGI, Modified Gingival Index; GBI, Gingival Bleeding Index; TMQH, Turesky modified Quigley–Hein plaque index; SE, standard error.

Authors, Year	Country/Setting/Population	Study Design (All Randomised Controlled Trials)	Intervention	Comparator	Outcome Measure(s)	Follow-Up Duration	Number of Participants (Dropouts in Brackets)	Results for Interproximal Plaque
Sjögren, K., Lundberg, A., Birkhed, D., Dudgeon, D. and Johnson, M., 2004 [21]	Sweden, adults	Four-period crossover design	Four regimens: 1. Manual brushing only with Jordan Active Tip toothbrush. 2. Manual brushing after flossing (as above, plus waxed floss from Johnson & Johnson). 3. Oral-B Ultra Plaque Remover (D9). 4. Sonicare Plus (now renamed Sonicare Advance)	See intervention	Wet weight inter proximal plaque (mg)	8 days for each regimen	47 (unclear if any dropouts)	Wet weight inter proximal plaque (mg) mean (SD) Day 1: OR brush = 14.8 (11.5); sonic brush = 5.7 (2.5) Day 8: OR brush = 17.2 (10.8); sonic brush = 6.0 (2.6)
Sharma, N., Goyal, C., Qaqish, J., Cugini, M., Thompson, M. and Warren, P., 2005 [22]	Clinic, adults	Randomised, examiner-blind, three-arm, single-use crossover design	OR/pulsating brush: Oral-B ProfessionalCare 7000, 340 Hz pulsation, 73 Hz oscillation	OR/pulsating brush: Oral-B 3D Excel (3DE), 340 Hz pulsation, 63 Hz oscillation; sonic brush: Sonicare Advance toothbrush, 260 Hz, Easy Start feature deactivated, used with “normal” settings	RMNPI, reduction safety	Minimum 8 days	79 (0)	Post-brushing RMNPI scores and plaque reduction (mean (SD)) PC7000: Post-brushing RMNPI = 0.33, (0.12) plaque reduction = 0.68 (0.12); 3DE: Post-brushing RMNPI = 0.32 (0.11), plaque reduction = 0.68 (0.11); sonic brush: Post-brushing RMNPI = 0.41 (0.15), plaque reduction = 0.59 (0.15)
Strate, J., Cugini, M., Warren, P.R., Qaqish, J., Galustians, H. and Sharma, N., 2005 [23]	Research facility, adults	Randomised, examiner-blind, crossover design with two study visits	OR toothbrush: Oral-B Professional Care Series, oscillation angle of 45 degrees, 73 Hz, with a pulsation frequency of 340 Hz	Sonic toothbrush: Sonicare Elite (260 Hz, Easy Start feature deactivated)	RMNPI, reduction safety	Minimum 4 days	61 (0)	Post-brushing RMNPI scores and mean plaque reduction (mean (SD)): OR toothbrush: Post-brushing = 0.034 (0.06), mean plaque reduction = 0.966 (0.06), % plaque removal = 96.6%; sonic toothbrush: Post-brushing = 0.271 (0.18), mean plaque reduction = 0.729 (0.18), % plaque removal = 72.9% *p* = 0.0001
Biesbrock, A., Bartizek, R., Walters, P., Warren, P., Cugini, M., Goyal, C. and Qaqish, J., 2007 [24]	Adults	Two studies used a randomised, examiner-blind, two-treatment, crossover design	Oscillating–rotating brush: Oral-B Triumph. This power brush has a round brush head and three-dimensional motion (rotation–oscillation plus pulsation). It operates at 8800 oscillations/40,000 pulsations per minute. The toothbrush was fitted with a FlossAction brush head.	Sonic brush: Sonicare Elite 7300. This power brush has a conventionally shaped brush head and side-to-side motion. It operates at a frequency of 260 Hz. The Easy Start feature was deactivated prior to use, and the brush was used with the normal settings. This toothbrush can be used with either the standard, full-size brush head or a newly designed compact brush head intended for smaller mouths and precision cleaning.	Whole-mouth plaque reduction (RMNPI) Marginal and interproximal plaque reduction (RMNPI)	Study 1: 2 weeks Study 2: Approximately 3 weeks	Study 1: 50 (0) Study 2: 49 (1)	Reduction in RMNPI score (mean ± SD) Study 1: OR brush = 0.8275 ± 0.07; sonic brush = 0.6323 ± 0.12 (OR brush 21% greater reduction than sonic brush, *p* < 0.001) Study 2: OR brush = 0.758 ± 0.007; sonic brush = 0.662 ± 0.007 (OR brush 14.6% greater reduction than sonic brush, *p* < 0.001)
Biesbrock, Walters, Bartizek, Goyal and Qaqish, 2008 [25]	Canada, adults	Randomised, examiner-blind, two-treatment, four-period, four-sequence crossover AABB, ABBA, BBAA, BAAB treatment sequences	Oscillating–rotating brush Oral-B Triumph with FlossAction brush head with MicroPulse bristles (8800 oscillations/40,000 pulsations per minute)	Sonic brush Sonicare FlexCare with ProResults brush head (260 Hz)	Full-mouth plaque score, RMNPI post-brushing	12 days approximately	48 (3)	Post-brushing plaque reduction adjusted mean ± SE: OR toothbrush: 0.884 ± 0.013 Sonic toothbrush: 0.724 ± 0.013
Putt, Milleman, Jenkins, Schmitt, Master and State, 2008 [26]	Independent clinical research organisation, Fort Wayne, Indiana, USA, adults	Single-use, examiner-masked, randomised, crossover	Sonic toothbrush: Sonicare FlexCare with ProResults radial brush head	Oscillating–rotating toothbrush: Oral-B Triumph Professional Care 9000 with FlossAction brush head	% reduction in full-mouth plaque score, Turesky modified Quigley–Hein plaque index (TMQH), plaque scores by site safety	2 weeks approximately	94 (1)	% reduction in TMQH for interproximal sites (mean ± SD): sonic toothbrush: 33.74 ± 14.79; oscillating–rotating toothbrush: 24.21 ± 14.27
Klukowska, M., Grender, J., Goyal, C., Mandl, C. and Biesbrock, A., 2012 [27]	Adults	Open-label, examiner-blind, two-treatment, parallel-group, randomised study	OR toothbrush: Oral-B Triumph with SmartGuide (aka Oral-B Professional Care SmartSeries 5000 in US) with EB25 FlossAction brush head	Sonic toothbrush: Philips Sonicare DiamondClean with DiamondClean standard brush head	MGI, GBI, RMNPI safety, consumer perception assessment	12 weeks	130 (0)	Adjusted mean reduction from baseline (SE), % reduction for interproximal RMNPI Week 6: OR toothbrush = 0.231 (0.0154), 23.3% Sonic toothbrush = 0.188 (0.0154), 19.1% Difference between brushes = 22.9%, *p* = 0.048 Week 12: OR toothbrush = 0.387 (0.0156), 31.9% Sonic toothbrush = 0.311 (0.0156), 31.6% Difference between brushes = 24.4%, *p* = 0.001
Klukowska, Grender, Conde and Goyal, 2013 [28]	USA? (unclear), adults	Randomised controlled, two-treatment, parallel-group, examiner-blind	Oscillating–rotating brush Oral-B Triumph with FlossAction AB25 brush head	Sonic brush Colgate ProClinical A1500 with Triple Clean brush head	MGI, whole-mouth RMNPI	12 weeks	130 (4)	Adjusted mean reduction from baseline (SE) Week 4: OR toothbrush: 0.275 (0.014); sonic toothbrush: 0.198 (0.014) Week 12: OR toothbrush: 0.407 (0.017); sonic toothbrush: 0.322 (0.016)
Klukowska, M., Grender, J., Conde, E., Ccahuana Vasquez, R. and Goyal, C., 2014 [29]	Research centre, adults	Randomised, two-treatment, examiner-blind, parallel-group design	OR toothbrush: Oral-B Triumph with SmartGuide with FlossAction brush head, D34/EB25	Sonic toothbrush: Sonicare FlexCare Platinum with InterCare standard brush head	MBI, GBI, number of bleeding sites, RMNPI safety	12 weeks	130 (3)	Interproximal RMNPI reduction (adjusted mean reduction (SE), % change 6 weeks: OR brush = 0.205 (0.0142), 20.7%; sonic brush = 0.181 (0.0141), 18.3%; 13.3% difference between brushes, *p* = 0.231 12 weeks: OR brush = 0.344 (0.0155), 34.7%; sonic brush = 0.309 (0.0153); 31.2% 11.3% difference between brushes, *p* = 0.112
Klukowska, M., Grender, J., Conde, E., Goyal, C., Qaqish, J. and Schneider, M., 2014 [30]	Unclear	Randomised, examiner-blind, two-treatment, parallel-group study	OR toothbrush: Oral-B Triumph with SmartGuide with Oral-B CrossAction brush head D34.EB50	Sonic toothbrush: Sonicare DiamondClean with standard brush head	MGI, GBI, number of bleeding sites, RMNPI safety, user experience	6 weeks	130 (2)	Six-week adjusted mean reduction from baseline RMNPI interproximal plaque (SE), % change OR toothbrush = 0.280 (0.0130), 28.4%; sonic toothbrush = 0.210 (0.0130), 21.3%; 33.3% difference between brushes, *p* < 0.001
Ccahuana-Vasquez, R., Conde, E., Grender, J., Cunningham, P., Qaqish, J. and Goyal, C., 2015 [31]	Unclear	Randomised, examiner-blind, two-treatment, parallel-group	Oscillating–rotating brush: Oral-B Professional Care 1000 [D16u] with Oral-B CrossAction brush head [EB50]	Sonic brush: Sonicare DiamondClean with standard DiamondClean brush head	RMNPI, MGI, GBI	8 weeks ± 2 days	150 (2)	Adjusted mean reduction (SE), % change (between baseline and week 8): OR toothbrush = 0.349 (0.0140), 34.9%; sonic toothbrush = 0.270 (0.0142), 27%; 29.3% difference between groups, *p* < 0.001
Lv, Guo and Ling, 2018 [32]	Guangzhou city, China	Randomised, examiner-blind, parallel-group, with two-treatment sub-trials	High-frequency sonic toothbrush: GEVILAN GET011 with GEH011 brush head (683 Hz) (Group A)	Oscillating–rotating toothbrush: Oral-B PRO 700 with Cross Action brush head (8800 oscillations and 20,000 pulsations per minute) (Group B) Sonic brush: Philips Sonicare Healthy White HX6712 with ProResults brush head (517 Hz) (Group C) MGI	RMNPI, MGI, GBI safety	6 months	120 (1)	Adjusted mean reduction from baseline (SE), % change Month 3: A: 0.114 (0.017), 11.5% B: 0.082 (0.017), 8.3% C: 0.108 (0.016), 11.1% Month 6: A: 0.169 (0.013), 17.1% B: 0.131 (0.013), 13.3% C: 0.219 (0.019), 22.4%
Adam, Goyal, Qaqish and Grender, 2020 [33]	Canada (unclear)	Randomised, open-label, parallel-group, examiner-blind	Oscillating–rotating brush Oral-B iO and Ultimate Clean brush head M7/OC15	Sonic brush Philips Sonicare DiamondClean Smart Sonic and Premium Plaque control brush head HX9903/11	Modified Gingival Index, Gingival Bleeding Index, RMNPI safety after using the assigned toothbrush for the study period	8 weeks	90 (0)	Adjusted mean (SE) change from baseline: Sonic: 0.324 (0.0234) Oscillating–rotating: 0.458 (0.0234)
Goyal, C., Adam, R., Timm, H., Grender, J. and Qaqish, J., 2021 [34]	Germany/research centre	Single-centre, examiner-blind, two-treatment, open-label, parallel-group, randomised study	OR brush: Oral-B iO with Ultimate Clean brush head (M7/OR015) in Daily Clean mode	Sonic brush: Sonicare DiamondClean with Premium Plaque Control brush head (HX9903/11) Clean mode with an intensity level of 3 (high)	MGI, GBI, RMNPI	6 months	110 (0)	Adjusted mean (SE) change from baseline interproximal RMNPI Day 1—single brushing: OR brush = 0.891 (0.0101); sonic brush = 0.794 (0.0101); 12% difference between groups, *p* < 0.001. Week 1—OR brush = 0.116 (0.0126); sonic brush = 0.063 (0.0126); 84.5% difference between groups, *p* = 0.003. Week 24—OR brush = 0.467 (0.0247); sonic brush = 0.371 (0.0247); 25.8% difference between groups, *p* = 0.007

**Table 5 healthcare-12-01035-t005:** Risk of bias based on Cochrane Risk of Bias 2 [37] chart (studies [20,21,22,23,24,25,26,27,28,29,30,31,32,33]).

Study	Randomisation	Assignment to Intervention	Adherence to Intervention	Missing Outcome Data	Measurement of the Outcome	Selection of the Reported Result	Overall Risk of Bias
Sjögren et al. (2004) [21]			Use of own toothbrush for washout periods, unclear which toothbrush this was				
Sharma et al. (2005) [22]			Used normal toothbrush during washout period				
Strate et al. (2005) [23]			Use of own toothbrush for washout periods, unclear which toothbrush this was				
Biesbrock et al. (2007) [24]							
Biesbrock et al. (2008) [25]			Use of own toothbrush for washout periods, unclear which toothbrush this was				
Putt et al. (2008) [26]							
Klukowska et al. (2012) [27]			OR group used SmartGuide				
Klukowska et al. (2013) [28]							
Klukowska et al. (2014a) [29]			OR group used SmartGuide				
Klukowska et al. (2014b) [30]			OR group used SmartGuide				
Ccahuana-Vasquez et al. (2015) [31]				Two dropouts, both from comparator group; study underpowered?			
Lv et al. (2018) [32]							
Adam et al. (2020) [33]							
Goyal et al. (2021) [34]			App use differed between groups				

Key: Green = low-risk; orange = some concerns; red = high-risk.

**Table 6 healthcare-12-01035-t006:** Time-wise comparison of OR and sonic toothbrushes. OR, oscillating–rotating; RMNPI, Rustogi Modified Naval Plaque Index; TMQH, Turesky modified Quigley–Hein plaque index.

Follow-Up Time	Interproximal Plaque Reduction	Reference
1 day	9.1 mg less remaining wet weight interproximal plaque for sonic toothbrush than OR toothbrush	Sjögren et al. (2004) [21]
	0.097 greater interproximal RMNPI reduction for OR toothbrush than sonic toothbrush	Goyal et al. (2021) [34]
4 days	0.237 greater interproximal RMNPI reduction for OR toothbrush than sonic toothbrush	Strate et al. (2005) [23]
1 week	0.053 greater interproximal RMNPI reduction for OR toothbrush than sonic toothbrush	Goyal et al. (2021) [34]
8 days	11.2 mg less remaining wet weight interproximal plaque for sonic toothbrush than OR toothbrush	Sjögren et al. (2004) [21]
	0.09 greater interproximal RMNPI reduction for OR toothbrush than sonic toothbrush (at up to 8 days)	Sharma et al. (2005) [22]
12 days	0.16 greater interproximal RMNPI reduction for OR toothbrush than sonic toothbrush	Biesbrock et al. (2008) [25]
2 weeks	0.1952 greater interproximal RMNPI reduction for OR toothbrush than sonic toothbrush	Biesbrock et al. (2007) [24]
	9.52% greater TMQH reduction for sonic toothbrush than OR toothbrush	Putt et al. (2008) [26]
3 weeks	0.096 greater interproximal RMNPI reduction for OR toothbrush than sonic toothbrush	Biesbrock et al. (2007) [24]
4 weeks	0.77 greater interproximal RMNPI reduction for OR toothbrush than sonic toothbrush	Klukowska et al. (2013) [28]
6 weeks	0.043 greater interproximal RMNPI reduction for OR toothbrush than sonic toothbrush	Klukowska et al. (2012) [27]
	0.024 greater interproximal RMNPI reduction for OR toothbrush than sonic toothbrush	Klukowska et al. (2014a) [29]
	0.07 greater interproximal RMNPI reduction for OR toothbrush than sonic toothbrush	Klukowska et al. (2014b) [30]
8 weeks	0.079 greater interproximal RMNPI reduction for OR toothbrush than sonic toothbrush	Ccahuana-Vasquez et al. (2015) [31]
	0.134 greater interproximal RMNPI reduction for OR toothbrush than sonic toothbrush	Adam et al. (2020) [33]
12 weeks/3 months	0.076 greater interproximal RMNPI reduction for OR toothbrush than sonic toothbrush	Klukowska et al. (2012) [27]
	0.085 greater interproximal RMNPI reduction for OR toothbrush than sonic toothbrush	Klukowska et al. (2013) [28]
	0.035 greater interproximal RMNPI reduction for OR toothbrush than sonic toothbrush	Klukowska et al. (2014a) [29]
	Between 0.026 and 0.032 greater interproximal RMNPI reduction for two different sonic toothbrushes than OR toothbrush	Lv, Guo and Ling (2018) [32]
6 months	Between 0.026 and 0.032 greater interproximal RMNPI reduction for two different sonic toothbrushes than OR toothbrush	Lv, Guo and Ling (2018) [32]
	0.096 greater interproximal RMNPI reduction for OR toothbrush than sonic toothbrush	Goyal et al. (2021) [34]

**Table 7 healthcare-12-01035-t007:** Time-wise meta-analyses of interproximal plaque reduction, comparison of OR and sonic toothbrushes. OR, oscillating–rotating.

Follow-Up Time	Interproximal Plaque Reduction	*p* Value; Heterogeneity
6 weeks	0.005 greater interproximal RMNPI reduction for OR toothbrush than sonic toothbrush	*p* = 0.0008; I^2^ = 32%
8 weeks	0.09 greater interproximal RMNPI reduction for OR toothbrush than sonic toothbrush	*p* < 0.00001; I^2^ = 51%
12 weeks	0.04 greater interproximal RMNPI reduction for OR toothbrush than sonic toothbrush	*p* = 0.0001; I^2^ = 79%
6 months	0.03 greater interproximal RMNPI reduction for OR toothbrush than sonic toothbrush	*p* = 0.09, I^2^ = 95%

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
