# Peer review of "The Effect of Different Electric Toothbrush Technologies on Interdental Plaque Removal: A Systematic Review with a Meta-Analysis"

_healthcare, 2024, doi:10.3390/healthcare12101035_

Round 1

Reviewer 1 Report

Comments and Suggestions for Authors

Dear Authors,

I have carefully reviewed your study examining the efficacy of toothbrushes with rotating and sonic technology for interproximal plaque removal in adult patients.

Upon my evaluation, I find the manuscript to be well-organized and in compliance with the requirements of a systematic review. In the Introduction section, I suggest that you incorporate a brief chapter discussing the relationship between oral cavity microbiota and systemic diseases, emphasizing the significance of finding an effective brushing technique, as demonstrated in studies such as doi:org/10.1111/prd.12393, doi:10.3390/jcm10173874.

Regarding the Results section, I recommend simplifying the description and refraining from repeating the information in the tables. Additionally, please simplify the description of the meta-analysis. In the Discussion section, I encourage you to present a one chapter with a with a single connected speech. I hope this feedback is helpful in enhancing your manuscript.

Author Response

Thanks a lot for your review and advice.

We have included some discussion on systematic conditions associated with oral health
We have simplified the results and meta-analysis
We have made the discussion one continuous section

Reviewer 2 Report

Comments and Suggestions for Authors

Dear authors, 

This is a very well-designed and conducted systematic review. The findings are interesting to the readers and well-written.

The systematic review by Lewis et al. reported on the efficacy of different toothbrush technologies in removing interdental plaque. The methodology of the systematic review is meticulous and followed the PRISMA guidelines and had clearly defined PICOS. The results from the 17 included studies in the review clearly demonstrated that oscillating-rotating toothbrushes removed more inter-proximal plaque than oscillating toothbrushes. The risk of bias of the included studies was assessed with Cochrane Risk of Bias 2 tool in which the majority of studies were deemed of low risk.

The conclusions are supported by the findings of the systematic review. The figures and tables appropriately summarize the findings. ‎Lastly the references are up to date and appropriate.‎ The manuscript is appropriately written with excellent level of English and fits within the scope of the ‎journal. The topic covered is very interesting to a wide range of oral healthcare providers.

I could advise the authors to rather include some of the tables (perhaps Table 1 or 2) as supplementary files rather than within the manuscript

Overall, I believe this manuscript is of very high quality and will be valuable to the readers. It should be accepted for publication.

Sincerely 

Author Response

Thanks a lot for your review and advice.

We are happy for tables 1&2 (now tables 2 and 4; study characteristics and time-wise comparison) to be supplementary files.

Reviewer 3 Report

Comments and Suggestions for Authors

Incorporate these feedback points into your manuscript to strengthen the presentation of findings and their implications. If you have any further questions or need clarification, feel free to reach out

  • Explicitly state the gap in the literature that your study aims to address. While you hint at the importance of comparing different electric toothbrush types, make this gap more explicit.
  • Engage more deeply with the literature by discussing the implications of previous findings and the gaps that remain in the field. This will strengthen the rationale for your study.
  • Address any limitations or challenges encountered during the meta-analysis (e.g., heterogeneity, study quality) and how these were managed in the interpretation of results.
  • While you mentioned using Cohen's Kappa score to assess inter-rater agreement, consider providing more context on the interpretation of this score. Describe the level of agreement achieved and how disagreements were resolved.
  • Use precise language to convey the level of certainty in the findings and recommendations in conclusion (e.g., "tendency towards increased reduction" rather than definitive statements).

Author Response

Thanks a lot for your review and advice.

We have gone into more detail on the gap in the literature
We have discussed more about the findings of the existing literature
We have discussed the limitations of the meta-analysis in relation to the conclusions
We have discussed the Kappa score in more detail
We have ensured “tendency towards reduction” has been used.

Reviewer 4 Report

Comments and Suggestions for Authors

VCXdear authors congratulations on your manuscript, it is very interesting but needs to be improved to merit publication.

The introduction is poor with articles that have dealt with the same subject in the past, please add more articles that have dealt with the same subject.

Make a chart with inclusion and exclusion criteria

Conclusions are poor, please expand the conclusions and explain them more comprehensively.

Author Response

Thanks a lot for your review and advice.

We have included 2 further systematic reviews in the introduction. 
We have inserted tables for the inclusion and exclusion criteria
We have expanded further on the conclusions.